# Wing structure and neural encoding jointly determine sensing strategies in insect flight

**Alison I. Weber** *, **Thomas L. Daniel**, **Bingni W. Brunton**

Department of Biology, University of Washington, Seattle, Washington, United States of America

* aiweber@uw.edu

**Data Availability Statement:** Code that reproduces all simulation data can be found at https://github.com/aiweber/optimal_sensing_ELwing.

**Funding:** This work was supported by the Washington Research Foundation (AIW); the

## Abstract

Animals rely on sensory feedback to generate accurate, reliable movements. In many flying insects, strain-sensitive neurons on the wings provide rapid feedback that is critical for stable flight control. While the impacts of wing structure on aerodynamic performance have been widely studied, the impacts of wing structure on sensing are largely unexplored. In this paper, we show how the structural properties of the wing and encoding by mechanosensory neurons interact to jointly determine optimal sensing strategies and performance. Specifically, we examine how neural sensors can be placed effectively on a flapping wing to detect body rotation about different axes, using a computational wing model with varying flexural stiffness. A small set of mechanosensors, conveying strain information at key locations with a single action potential per wingbeat, enable accurate detection of body rotation. Optimal sensor locations are concentrated at either the wing base or the wing tip, and they transition sharply as a function of both wing stiffness and neural threshold. Moreover, the sensing strategy and performance is robust to both external disturbances and sensor loss. Typically, only five sensors are needed to achieve near-peak accuracy, with a single sensor often providing accuracy well above chance. Our results show that small-amplitude, dynamic signals can be extracted efficiently with spatially and temporally sparse sensors in the context of flight. The demonstrated interaction of wing structure and neural encoding properties points to the importance of understanding each in the context of their joint evolution.

## Author summary

In addition to generating forces for flight, insect wings also serve an important role as sensory structures, providing rapid feedback about wing bending that is used to stabilize flight. While much is known about how wing structure affects aerodynamic performance, the effects of wing structure on sensing remain unexplored. Using a computational model of a flapping wing, we examine how sensing strategies depend on wing stiffness and sensor properties. We show that body rotations can be accurately detected with a small number of sensors on the wing across a wide range of conditions. Optimal sensor locations are clustered at either the wing base or wing tip, depending on a combination of wing stiffness and sensor properties. Moreover, sensing performance is robust to multiple kinds of

eScience Institute at the University of Washington (AIW); and the Air Force Office of Scientific Research awards FA9550-18-1-0114 (BWB) and FA9550-19-1-0386 (BWB, TLD). The funders had no role in study design, data collection and analysis, decision to publish, or preparation of the manuscript.

**Competing interests:** The authors have declared that no competing interests exist.

perturbations. Our work provides a basis for understanding how wing structure impacts incoming sensory information during flight.

## Introduction

The physical structure of an animal's body transforms the incoming sensory information and can either facilitate or constrain sensing capacity. Indeed, body parts in many systems serve to preprocess sensory inputs in ways that are beneficial for the organism, extracting relevant features and reducing downstream computational burdens. For instance, the decrease in stiffness from the base to apex of the mammalian cochlea promotes frequency selectivity along its length [1], and the response properties of mechanosensors that encode high-frequency vibrations in mammalian skin are determined largely by the viscoelastic properties of layered, fluid-filled capsule surrounding nerve endings [2]. In systems that rely on mechanosensation in particular, there is a large body of work pointing to the importance of structure in preprocessing sensory inputs (reviewed in [3]). On the other hand, there may be structural limits that constrain the numbers and locations of sensory receptors. In the vibrissal system, for example, mechanosensory neurons are located only at the whisker base, so information about an object's point of contact with the whisker cannot be directly measured [4]. Thus, the physical properties of non-neural structures play an important role in determining how stimuli are experienced and transduced by sensory receptors embedded in the body.

Sensory receptors transduce stimuli into electrical signals, typically taking the form of action potentials, in which the incoming signal is converted into a series of all-or-none events. Sensory neurons respond selectively to particular features of the stimulus; for instance, auditory neurons in the cochlea respond to particular frequencies, and visual neurons in the retina respond to distinct temporal patterns of illumination [5–7]. This transformation from input to response can often be accurately represented by a two-part model of neural encoding [8, 9]. One part consists of a stimulus feature or features to which the neuron is sensitive, and the second part is a nonlinear function that captures how selective the neuron is for those features. Importantly, neural encoding properties define the information available to the rest of the nervous system about the animal's environment and its own body. These properties may adapt across multiple timescales, changing over rapid timescales based on the history of inputs as well as over evolutionary time [10, 11].

In this paper, we focus on the interaction of body structure and sensory encoding properties in the context of insect flight control, where wings provide rapid sensory feedback necessary for stable flight [12–16]. Although extensive previous work has examined how wing structure impacts aerodynamic performance (for a few examples, see [17–22]), the impacts of wing structure on sensing remain unexplored. Structural properties (e.g., geometry, flexural stiffness) interact with forces acting on the wing to produce local spatiotemporal patterns of strain that are sensed by the nervous system. Local strain is encoded by sensors, called campaniform sensilla, distributed sparsely over the wings at consistent locations across individuals of a given species. It has recently been demonstrated that only a small number of sensors are needed to read out behaviorally relevant information about body rotations, with neural sensors providing advantages over sensors that directly encode strain [23]. Sparse sensing strategies have advantages in terms of both energetic cost and robustness [24–27]. However, no previous work has examined the impact of wing structure on sparse sensing strategies.

In this paper, we show that just a few spiking sensors can detect subtle differences in strain that arise from the relatively tiny forces produced by body rotation compared to wing flapping.

We use a simple model, with wing size and stiffness based on the wings of *Manduca sexta*, to characterize spatiotemporal patterns of strain over a wing during flapping. We then encode this strain in a population of spiking sensors, which we refer to as *neural-inspired sensors*, whose response properties are based on experimental measurements from mechanosensory neurons. With a sparsity-promoting optimization method, we solve for the locations of a small, fixed number of sensors for detecting body rotation about different axes. We focus on systematically varying wing stiffness because it is an important structural property that varies widely across species [28], within species [29], and even over an animal's lifetime [30]. We demonstrate that wing stiffness and neural encoding properties jointly determine the accuracy of body rotation detection and optimal sensor locations. Moreover, sensing performance is remarkably resilient to sensor loss and external disturbances, suggesting that sparse placement of sensors can be used for efficient, robust sensing in the context of flight.

## Results

To understand how sensing strategies and performance are impacted by wing structure, we focus on one specific sensing goal: finding a minimal set of strain-encoding, spiking mechanosensors on the wing that are effective in detecting body rotation. Next, given the timing of a single spike from each of a set of sensors placed at optimal locations, we ask how well rotation of the body about different axes can be discriminated. We then analyze this performance and the optimal locations of the sensors on the wings, assessing how they change as we manipulate wing stiffness and neural encoding properties.

We first simulate spatiotemporal patterns of strain over a flapping wing using an Euler-Lagrange model inspired by the wings of the hawkmoth *Manduca sexta* [31] (Fig 1A, top). The dimensions, flexural stiffness, and flapping frequency are based on previously measured quantities [29]. Strain data are acquired for two conditions: one in which the wing is flapping, and another in which the body is undergoing rotation while the wing is flapping. The time history of strain is then encoded by a population of neural-inspired sensors, whose encoding properties are based on experimental measurements of mechanosensors in hawkmoth wings [14]. The output of this neural encoding step is a pattern of spikes, temporally sparse all-or-none signals (Fig 1A, middle). We then solve for a spatially sparse set of sensors (10 sensors, unless otherwise noted) that can be used to detect body rotation based on spike timing (Fig 1A, bottom). A single spike time from each sensor in this subpopulation is used to determine whether or not the wing is undergoing rotation during each wingbeat. We repeat this procedure for wings of different stiffness (Fig 1B, top) and sensors with different thresholds (Fig 1B, bottom).

### Spike timing and precision determine local peaks in accuracy as a function of wing stiffness

To examine how sparse sensing strategies depend on wing structure, we first vary wing stiffness while holding neural encoding properties constant (linear filter frequency parameter $\omega = 1$; neural threshold $\beta = 0.2$). The range of wing stiffness values examined centers on the average reported flexural stiffness for hawkmoth forewings (Methods) [28, 29]. We solve for sparse sensor locations that can detect rotation and evaluate performance using the 10 best sensors.

We begin by focusing on rotation in the yaw axis and later turn to pitch and roll. Classification accuracy changes nonmonotonically with wing stiffness, peaking near 100% accuracy at a wing stiffness similar to that for hawkmoth wings (stiffness factor = 1). There is a second, smaller peak of approximately 75% accuracy at a much lower stiffness (Fig 2A). In both cases, the optimal sensor locations occur at the wing tip. The single best sensor location for each of

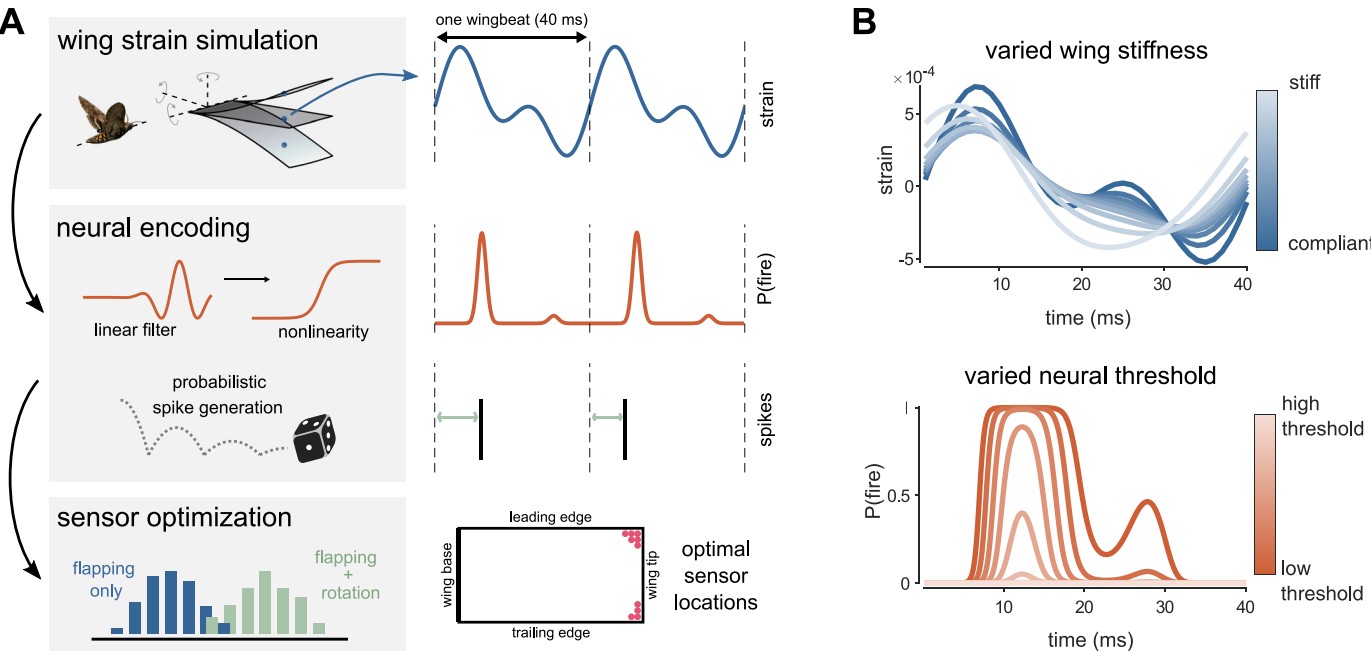

**Fig 1. Wing stiffness and neural encoding properties determine optimal sensor locations on a flapping wing.** A: *Top*: Strain over time is simulated at each possible sensor location in an Euler-Lagrange model of a flapping wing [23, 31]. The wing may be subject to rotation in different axes. Strain is shown for an example sensor location (blue dot). *Middle*: Strain at each location is encoded by a neural-inspired sensor. Strain is convolved with a linear filter and passed through a static nonlinearity to generate a probability of firing a spike over time, P(fire). Spikes are probabilistically generated, so that exact spike time varies from wingbeat to wingbeat. *Bottom*: The full data set, a matrix of spike times at all sensor locations for each wingbeat, is used to find a subset of sensors from which information about rotation can be read out. Linear discriminant analysis is used to find the vector *w*, such that when the data are projected onto *w* the means of the flapping only and flapping with rotation data are maximally separated. B: *Top*: Changing wing stiffness produces different strain (shown for identical location) over the course of each wingbeat. *Bottom*: Changing the threshold of the neural encoding nonlinearity alters of the probability of firing over the course of the wingbeat, with higher thresholds resulting in lower probability. Hawkmoth image in A courtesy of Armin J Hinterwirth.

these stiffness values is shown in Fig 2A, although all of the 10 best sensors in each case fall at the wing tip and show similar patterns of strain and spiking over time.

Differences in spike timing and spike timing precision underlie these local peaks in accuracy. For more compliant wings, the filtered strain in optimal sensors quickly crosses threshold (Fig 2B, left), leading to sharp transitions from zero probability of firing (P(fire)) a spike to certainty of firing a spike (P(fire) ∼1, Fig 2C, left). This results in precisely timed spikes both when the wing is flapping only and flapping with rotation (Fig 2D and 2E, left). Small differences in filtered strain under these two conditions translates to small but detectable differences in spike timing. With as few as 10 sensors, wing rotation can be detected with ∼75% accuracy.

For stiffer wings, the filtered strain in optimal sensors varies less over time, barely reaching threshold (Fig 2B, right). Thus, there is an overall lower probability of firing, but the difference between flapping only and flapping with rotation is amplified: small differences in filtered strain translate to clear differences in P(fire) over time (Fig 2C, right). This change in P(fire) is reflected in spike timing, with the first spike being much more likely to occur earlier in the wingbeat cycle in the flapping case (around 5 ms) compared to flapping with rotation (around 25 ms; Fig 2D and 2E, right). Lower precision in spike timing is offset by large differences in the time to the first spike, resulting in a classification accuracy approaching 100%.

These results suggest that classification performance for a given wing stiffness might be improved by altering the neural threshold. We next examine how neural encoding properties

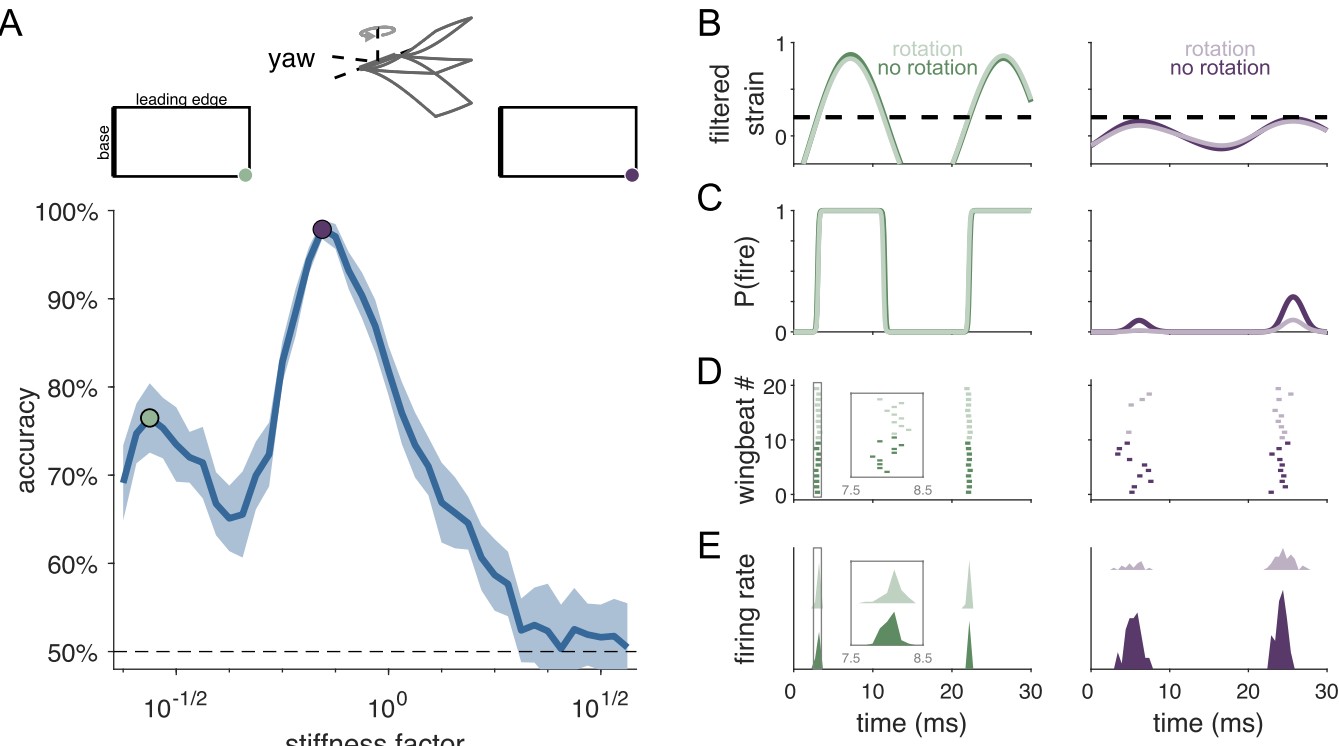

**Fig 2. Wing stiffness interacts with neural threshold to determine spike timing.** A: Accuracy as a function of wing stiffness. Chance accuracy is 50%, and stiffness factor 1 corresponds to 3 GPa, comparable to experimentally measured stiffness in the hawkmoth [28]. Green and purple dots indicate local peaks in accuracy. Shaded area indicates ±1 standard deviation. *Top*: Center schematic indicates the direction of rotation. Left and right schematics show the location of the single best sensor for each stiffness value. B: Filtered strain over a single wingbeat for flapping only ("no rotation") and flapping with rotation ("rotation") conditions at a single sensor location. Green and purple sensor locations in A correspond to left/green and right/purple filtered strain. Dashed horizontal line indicates the threshold (value at half-max) of the neural encoding nonlinearity. C: Filtered strain passed through the nonlinear function gives probability of spiking over time, P(fire). D: Spiking responses for 10 wingbeats each of the no rotation and rotation conditions. Spikes are generated probabilistically from P (fire), resulting in variable spike timing from wingbeat to wingbeat. *Inset*: Magnified view of the first firing event, surrounded by gray box, between 7.5 and 8.5 ms. E: Histogram of spike times (PSTHs) for each condition, summarizing spike timing over hundreds of wingbeats.

interact with wing stiffness by simultaneously varying both wing stiffness and the threshold of the nonlinearity in neural encoding.

## Structure and neural encoding interact to determine accuracy and optimal sensor locations

The neural threshold determines how selective the sensor will be for the feature given by the linear filter, with higher thresholds imparting stronger selectivity. The slope of the nonlinearity most strongly affects spike timing precision, with higher slopes leading to faster transitions from P(fire) = 0 to 1 and therefore greater spike timing precision. We set the slope of the non-linearity to roughly match the spike timing precision observed in wing mechanosensors (0.1–1 ms, [13]). The frequency content of the linear filter is chosen based on experimental observations [14]. Higher-frequency filters show uniformly low performance as a function of wing stiffness, while lower frequency filters show uniformly high performance (Fig A in S1 Appendix).

The ability to detect body rotations depends jointly on wing stiffness and neural threshold, with very dissimilar combinations yielding similarly high classification accuracy (Fig 3A). For

 Wing structure and neural encoding jointly determine sensing strategies in insect flight

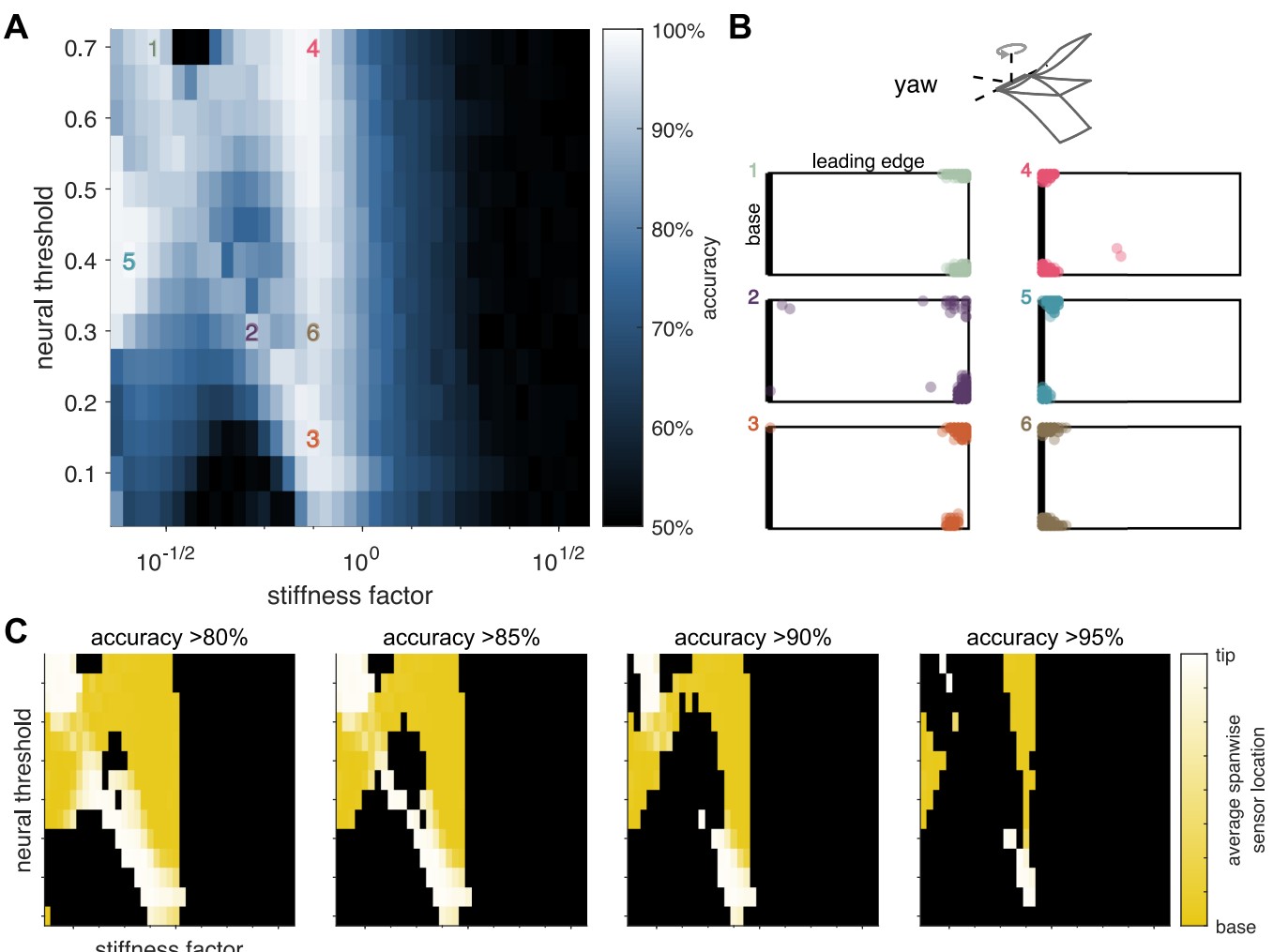

**Fig 3. Wing stiffness and neural threshold interact to determine sensor locations and classification accuracy in the yaw axis.** A: Accuracy as a function of neural threshold and wing stiffness. B: Optimal sensor locations of 10 best sensors, overlaid for 20 different simulated data sets. Neural threshold and wing stiffness of each panel are indicated by the corresponding colored dot in A. *Top*: Schematic indicates the axis of rotation. C: Optimal sensor locations in the spanwise direction from wing base (yellow) to wing tip (white) as a function of neural threshold and wing stiffness. Each panel shows results for a different accuracy cutoff. Black indicates parameter combinations that fall below the cutoff.

example, for stiffness factors just below 1, accuracy is similarly high for low (0.15) and high (0.7) thresholds, but accuracy is lower at intermediate values (e.g., 0.3).

Moreover, the optimal sensing strategies for a given wing stiffness can be qualitatively different for high and low thresholds. At stiffness factors just below 1, optimal sensors are located at the corners of the wing tip for low thresholds (Fig 3B, orange) and at the corners of the wing base for high thresholds (Fig 3B, pink). Note that sensors are expected to fall at the wing corners, as observed in previous work [23], because rotation elicits a twisting mode in wing bending, which leads to the largest differences in strain at wing corners between rotating and non-rotating conditions [31]. However, the distinction between sensors located at the wing base and those at the wing tip was not seen in previous work and is not necessarily expected [23]. When the spanwise sensor location (wing base to wing tip) is plotted as a function of neural threshold and wing stiffness, distinct regions of qualitatively different encoding strategies

emerge (Fig 3C). Interestingly, these regions have rather convoluted, discontinuous shapes, illustrating that the optimal sensing strategy reflects the complex interaction of structural properties and neural encoding properties. Sensing accuracy and sensor locations are minimally impacted if sensors are allowed to be located on two identical wings, one on each side of the body (Fig B in S1 Appendix).

## For different classification tasks, a simple encoding principle emerges

We next perform a similar set of numerical experiments for detecting rotation in each of the roll and pitch axes. Classification accuracy is generally lower for pitch detection compared to roll and yaw, with maximum accuracy across all parameter combinations tested only 77% (Fig 4). Several clear differences emerge between classification of roll and pitch compared to yaw. First, for roll and pitch, classification accuracy generally increases with increasing wing stiffness, whereas accuracy for yaw generally decreases with increasing wing stiffness. Second, optimal sensors are distributed along the wing base for roll and pitch for nearly all parameters tested (Fig 4B and 4D). This observation is consistent with the fact that twisting modes elicited by pitch and roll are much smaller or nonexistent, respectively, compared to yaw. For pitch and roll, there is a far more straightforward relationship between wing stiffness and accuracy, with little dependence on neural threshold. A single encoding strategy emerges, with optimal sensors at the wing base.

Thus far, we have considered rotation about three orthogonal axes individually. However, an animal potentially needs to detect rotation about multiple axes, or rotation about axes intermediate to primary body axes. We assessed optimal sensing strategies in each of these cases and found that they reflect a combination of the strategies used for single-axis detection, rather than fundamentally different strategies (Figs D, E, and F in S1 Appendix).

## Robustness to perturbations

To determine the extent to which these results are robust to different perturbations, we test two types of perturbations that reflect likely disruptions to both biological and engineered systems: sensor loss and external disturbances. Sensor loss may arise from wing damage sustained over an animal's lifetime or due to sensor failure in engineered systems. We select two wing stiffness/neural threshold parameter combinations that yield near-peak accuracy (∼100% for yaw and roll rotation; ∼75% for pitch rotation). For yaw axis rotation, one parameter set results in sensors placed at the wing base and one at the wing tip; for pitch and roll, both are located at the wing base. We randomly eliminate between 1 and 9 out of 10 sensors and compute accuracy using the remaining sensors. Accuracy falls smoothly as sensors are randomly dropped with, surprisingly, a single sensor still providing ∼75% accuracy on average for yaw and roll, and ∼ 60% accuracy for pitch (Fig 5A, 5B and 5C). Accuracy falls more gradually for pitch detection, likely due to the lower starting accuracy.

We next examine the effects of external disturbances to wing rotation, which may arise from environmental fluctuations such as wind gusts. For yaw, pitch, and roll detection, accuracy falls smoothly as a function of the disturbance magnitude (Fig 5D, 5E and 5F). The details of how accuracy falls depend on the specific pairing of nonlinear threshold and wing stiffness: while accuracy typically decreases gradually with increasing disturbance, one case (low threshold for roll detection) exhibits a sharper drop in performance. Nevertheless, in all cases, accuracy remains well above chance for disturbances that are large relative to the magnitude of rotation, even when the standard deviation of the disturbance is the same magnitude as the rotation to be detected.

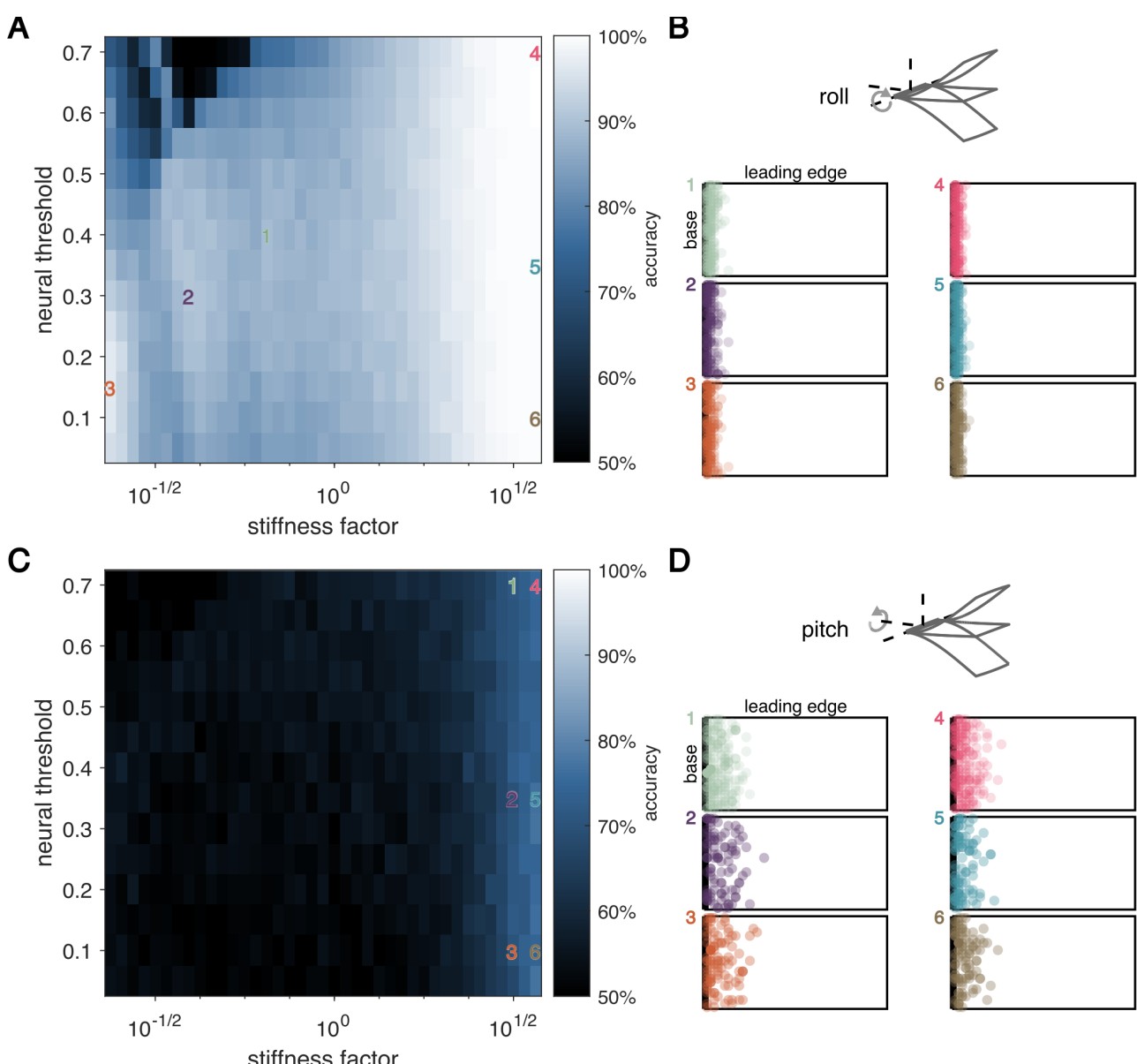

**Fig 4. For detecting rotation in the roll and pitch axes, sensor locations do not depend on stiffness or neural encoding properties.** A,C: Accuracy as a function of neural threshold and wing stiffness for roll and pitch, respectively. B,D: Optimal sensor locations of 10 best sensors, overlaid for 20 different data sets. Neural threshold and wing stiffness of each panel are indicated by the corresponding colored number in A. *Top*: Schematics indicate the axis of rotation.

In summary, accuracy falls smoothly as the strength of a perturbation increases, with no indication of catastrophic failures at some threshold level of perturbation. Moreover, the trends are nearly identical for very different sensors, even when those sensors are located on opposite ends of the wing and often when they have very different neural thresholds. We therefore see no evidence of systematic differences in robustness between sensors at different locations or with different thresholds.

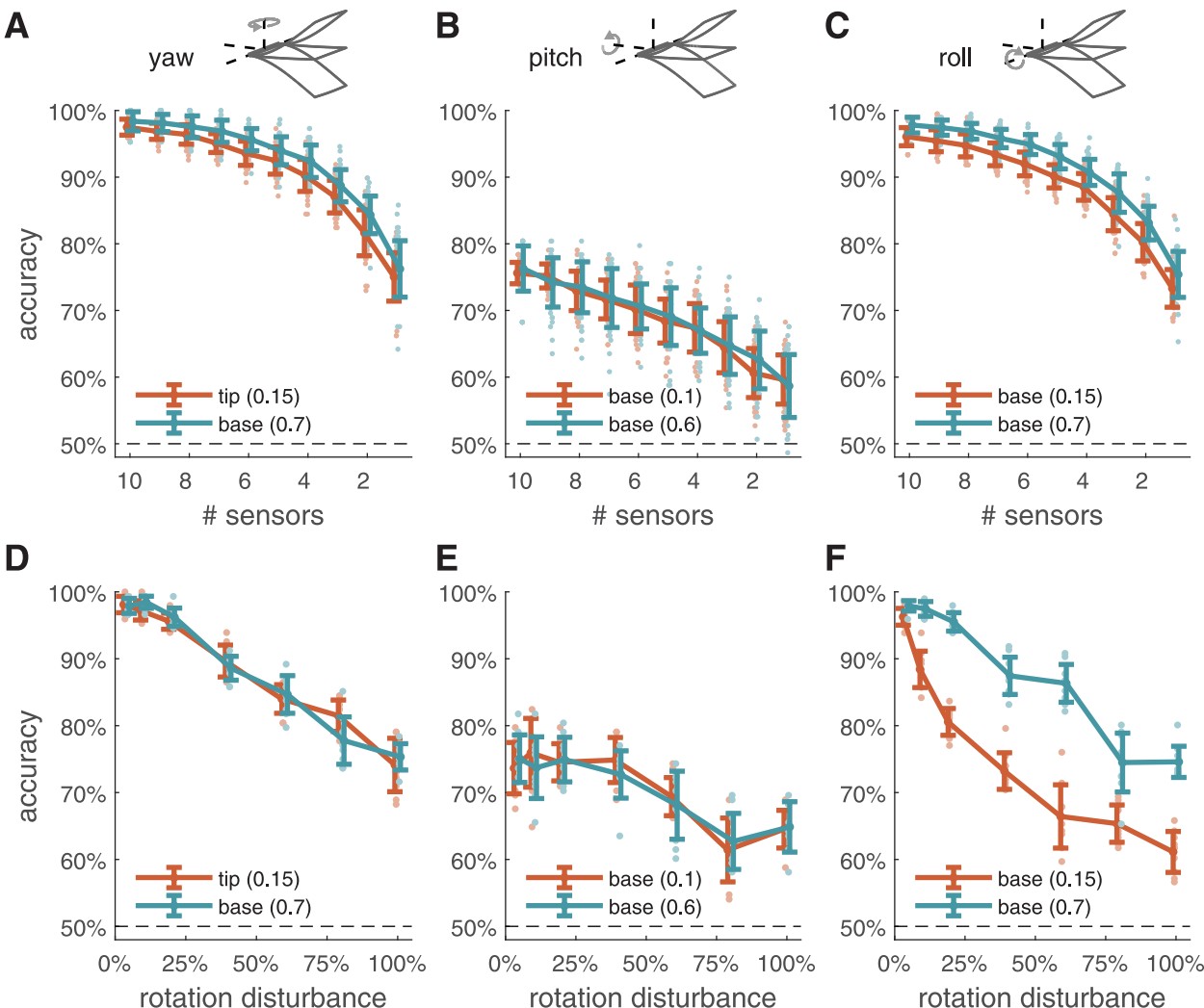

**Fig 5. Robustness to sensor dropout and external disturbance.** A: Accuracy of detecting rotation in the yaw axis as a function of number of sensors for two different wing stiffness/neural threshold parameter combinations (orange and teal). One parameter combination results in sensors at the wing tip, and a second results in sensors at the wing base, both with accuracy of 98% for 10 sensors. B–C: Same as A, for pitch and roll, respectively. In each case, two different parameter combinations were again chosen that produce similar accuracy with differing neural thresholds. For pitch-axis and roll-axis rotation detection, all sensors are located at the wing base. D–F: Accuracy as a function of the standard deviation of rotation disturbance for yaw, pitch, and roll, respectively, shown as the percentage of the constant rotation velocity to be detected. Parameter combinations same as corresponding A–C panels. Error bars in all panels denote ±1 standard deviation.

## Discussion

Our work is a first step towards understanding how insect wing structure determines incoming sensory information and thus sensing strategies. We examine the impacts of wing structure, in particular wing stiffness, on sensory encoding in a computational model of a flapping wing. We show that a small number of sensors, conveying spatially and temporally sparse signals can be used to reliably detect rotation over a range of wing stiffness values. Optimal encoding properties of these sensors and their placement vary with both wing stiffness and the need to identify rotation in different axes. Sensors are clustered at either the wing base or wing tip, depending on a combination of these factors. Moreover, accuracy is robust to multiple

types of perturbations, and robustness does not depend on the particular details of sensor placement. The interaction of wing structure and neural encoding properties points to the importance of considering how these features may evolve together to enable sensing.

## Optimal sensing strategies for flight in engineered and biological systems

Designing sensing strategies—including sensor number, placement, response properties, and sensor readout—has long been of interest to engineers who seek to build efficient, lightweight, low-cost systems. Much work in the domain of flight has focused on sensing in fixed-wing air-craft, rather than in flapping flight as exhibited by flying animals [32–36]. In a similar vein, these principles of optimality, efficient sensing, sparsity, and cost reduction have also informed neuroscientists' understanding of the nervous system [26, 37–40]. In many cases, the encoding properties of sensory neurons are matched to the statistics of the incoming signals. These encoding properties may be tuned over many timescales, from evolutionary time, to dynamically on subsecond timescales to match changing statistics in the environment [41, 42]. Therefore, understanding optimal sensing strategies provides a useful benchmark for understanding biological systems.

Prior studies have considered optimal strain sensing strategies in insect wings [23, 43–46]. Although using different sensing tasks and optimization criteria, taken together, they show that remarkably few sensors are needed to detect or identify patterns of strain on the wing. However, all of the previous work assumes sensors that either directly sense strain or encode strain as a continuous variable. In contrast, our current work uses spiking sensors. Spiking sensors are not only more faithful to the corresponding biological systems, where primary sensory neurons on the wing encode information in all-or-none action potentials, but they also enforce that sensors transmit temporally sparse signals. Investigating spiking sensors could present opportunities for more efficient, bio-inspired sensing approaches in engineered systems as well.

## Strain sensor placement in insect wings

Insect wings exhibit widely varying patterns of mechanosensor placement on their wings [47–51]. Even so, we have little understanding of the functional consequences of this diversity. Moreover, comparison between species is complicated by the vast differences in wing mor-phology and stiffness [29, 52]. Despite these differences, some generalizations can be made about the placement of strain sensors (campaniform sensilla). Perhaps the most striking com-monality across diverse taxa is the relative abundance of campaniform sensilla near the wing base [47–49, 51, 53–55]. Taken together, the results in the present work suggest that clustering sensors at the wing base may serve as a catch-all strategy, favorable across a wide range of wing and neural encoding properties to detect rotations about multiple axes (Figs 3 and 4 and Fig F in S1 Appendix).

## Additional features driving optimal sensor placement

Insect species vary widely in features that are likely to play a role in determining effective mechanosensor placement on the wings: wing size, shape, and venation pattern, among others. Understanding the functional consequences of these different features and how they impact spatiotemporal patterns of strain on the wing will be necessary for elucidating principles guid-ing the differing placement of sensory structures across diverse insect taxa. Answering these questions will require different computational models that allow more flexible manipulation of these variables. For example, models based on the finite element method (FEM) support more realistic wing geometry, wing venation patterns, and non-uniform stiffness [29, 49]. The

consequences of many of these properties have been previously explored in a variety of models, although none from a sensing perspective [18, 19, 29, 56, 57]. More biologically realistic features will result in more complex spatiotemporal patterns of wing strain, likely driving sensor placement to be far more complex and varied than those uncovered in the current study.

The wingstroke trajectory and kinematics may play a particularly important role in determining optimal sensor placement. In the current work, we see that the wingstroke, modeled as a sum of sines as in previous work [23, 31], drives two distinct response events over the course of single wingbeat (Fig 2). Interestingly, these response events interact with spike timing precision in sometimes counterintuitive ways, such that lower spike timing precision can sometimes yield greater discriminability. More complex wingstroke kinematics likely play an important role in determining the phase (or phases) within a wingstroke when each neuron is most sensitive. The trajectory of the wingstroke may also be used as a mechanism for active sensing, in which the animal modifies its motor output to acquire more useful sensory information [58–62]. Understanding the consequences of varying motor output for incoming sensory information will be crucial to understanding sensing strategies in flight.

Although we varied neural encoding properties (nonlinear threshold and filter frequency; Fig 3 and Fig A in S1 Appendix), all sensors on a given wing have identical encoding properties. Previous work suggests that there are likely two sub-populations of strain-sensitive neurons on the wings of moths [63]. It is possible that a wing with different sensor types may need fewer sensors and adopt a qualitatively different encoding strategy. Additionally, selectivity to multiple features has been demonstrated in neurons in multiple sensory systems, including strain-sensitive neurons in the halteres of crane flies [6, 64]. Allowing individual sensors to respond to multiple stimulus features may also allow for fewer sensors and different sensor positioning. However, it has recently been suggested that all strain-sensitive neurons on insect wings may, in fact, respond similarly in the context of flight (i.e., for rapidly fluctuating stimuli), suggesting that sensor placement may be the primary determinant of information encoded by a given sensor [65].

The current work investigates sensing strategies by finding solutions to a set of optimization tasks. However, biological systems need not operate at any true optimum, but may instead simply need to be "good enough" or robust to failure [66]. Solving for optimal sensor locations likely explains some of the features of the present study, for example, the abrupt shift between wing base and wing tip sensing strategies for yaw detection (Fig 3). If sensor locations are restricted to either the distal or the proximal half of the wing, sensor performance does not fall off sharply at these same transitions (Fig G in S1 Appendix). We investigated robustness of the optimal sensing strategies in the current study and found little difference between different sensor properties (locations or threshold), suggesting that a need for robustness would not favor one of these strategies over another. Exploring the range of sensing strategies that meet some (sub-optimal) performance criterion may be an interesting area of future work.

## Sensing needs are one of many evolutionary demands on wing structure

An organism's evolution reflects a combination of evolutionary constraints and an organism's needs [67, 68]. For insect wings, these needs may include force generation for flight, sensing, mating display, and predator avoidance, among others [69–71]. How wing structure impacts aerodynamic performance has been extensively studied [17–20, 22, 72–74]. These studies generally point to the importance of stiffness gradients (from base to tip and from leading edge to trailing edge) and rigidity provided by wing veins. However, the consequences of wing structure for sensing have remained unexplored, despite the fact that insect wings have long been known to provide sensory feedback during flight [12, 75, 76].

Incorporating additional constraints may shed light on how these needs interact. For example, in insects, campaniform sensilla on the wings are generally restricted to be on or near wing veins [47, 53, 63, 77]. Because we used a highly simplified wing model that did not include wing veins (or any spatial variation in stiffness), we allowed sensors to be placed at any point on a dense grid over the surface of the wing. Constraining sensor locations is straightforward and can be achieved by incorporating an additional penalty in the optimization step to decrease the likelihood of sensors being located in particular regions.

In the present work, we focus on how wing structure affects sensing and do not explore how sensing might interact with other roles of the wing, such as actuation, predator avoidance, and mating display. An important, and challenging, open question is: how are these diverse needs balanced over evolutionary timescales? Answering this question demands an interdisciplinary approach that combines biomechanics, neuroscience, and evolutionary biology to produce a more complete understanding of the processes driving wing evolution.

## Methods

Code that implements strain simulations, conversion of strain to spiking data, and sensor optimization can be found at https://github.com/aiweber/optimal_sensing_ELwing.

### Euler-Lagrange simulations of wing strain

We first simulate spatiotemporal patterns of strain over the surface of the wing using a previously developed Euler-Lagrange model based on parameters of wings of the hawkmoth *Manduca sexta* [31]. In this model, the wing is represented by a flat plate with a span of 50 mm, chord length of 25 mm, and thickness of 0.127 mm. Following prior modeling efforts [31], wing flapping motion is modeled as a sum of sines, with a primary frequency of 25 Hz and a secondary frequency of 50 Hz (76% amplitude). The wing is subject to rotation in different axes of comparable magnitude to rotations experienced by a hawkmoth during free flight (constant at 10 rad/s unless otherwise noted) [78]. Additionally, noise is added to both the flapping velocity as well as the rotation rate to simulate noise experienced during flapping flight, as in previous work [23]. Unless otherwise noted, the magnitude of noise in rotation is 1% of the rotation rate, and the magnitude of noise in flapping is 2% of the flapping amplitude. For the roll detection task, 2% noise in flapping is included with no additional noise in rotation, as rotation occurs in the same axis as flapping. For the four-way classification task, noise in all axes is included in all conditions.

We vary Young's modulus to explore the effects of changing structural properties of the wing, namely wing flexural stiffness. The range of explored modulus values is approximately centered at 3 GPa (stiffness factor = 1), corresponding to a spanwise flexural stiffness of $\sim 1.5 * 10^{-4}$ Nm$^2$, comparable to average experimentally measured spanwise flexural wing stiffness in *Manduca sexta* [28]. We vary Young's modulus values from 0.7 GPa to 10.0 GPa (flexural stiffness $3.6 * 10^{-5}$ to $5.6 * 10^{-4}$ Nm$^2$). This approximately corresponds to the flexural stiffness range from half the wing span to the wing base [29]. Below 0.7 GPa, simulations did not achieve appropriate levels of numerical convergence, and more flexible wings therefore cannot be tested with this model.

We simulate strain over a dense grid representing sensor locations with 1 mm spacing, resulting in 26 chordwise and 51 spanwise sensors, for a total of 1,326 sensor locations. Unless otherwise noted, we use normal strain in the spanwise direction. This provided improved performance over normal strain in the chordwise direction (Fig H in S1 Appendix). See [31] for additional details of Euler-Lagrange simulations and [23] for additional details of simulations

with disturbances. We typically simulate 3 seconds of data (75 wingbeats) for each condition at a sampling rate of 10 kHz.

## Strain encoding with neural-inspired spiking sensors

Neural-inspired sensors encode strain following a procedure similar to earlier work [23]. At each sensor location, strain is first convolved with a linear filter, representing the temporal feature of strain which the sensor is most sensitive to. The filtered strain signal reflects the similarity between linear filter (i.e., feature) and the strain experienced by the sensor over time. Filtered strain is then transformed by a static nonlinearity, which determines the sensor's selectivity for that particular feature: sensors with high threshold will only respond when the temporal pattern of strain is very similar to the feature given by the linear filter, corresponding to strong selectivity. The output of the linear-nonlinear encoding represents a probability that the neuron will generate a spike.

The shapes of the linear filter and nonlinearity are based on previous electrophysiological recordings of responses in mechanosensors of the wing nerve [14]. The linear filter $f$ is defined as a decaying sinusoidal function:

$$f(t, \omega, \tau, \delta) = \cos(2\pi\omega(t + \tau)) \cdot \exp\left(\frac{-(t + \tau)^2}{\delta^2}\right),$$

where $\omega$ is the frequency of the filter, $\tau$ is the time offset to the peak, and $\delta$ is the decay time. In this work, $\omega = \frac{1}{2\pi}$ ms$^{-1}$, $\tau = 5$ ms, and $\delta = 4$ ms. The static nonlinearity (also called the nonlinear decision function or nonlinear activation function) is given by:

$$N(g_t, \alpha, \beta) = \frac{1}{1 + \exp(-\alpha(g_t - \beta))},$$

where $g_t$ is the filtered stimulus at time $t$, $\alpha$ is the slope parameter, and $\beta$ is the threshold parameter, where the function reaches half-maximum. We hold $\alpha$ constant at 50 and vary the threshold $\beta$. As in previous work, $g$ is normalized by a constant (identical for all sensors) such that the maximum value of $g$ over all sensors is approximately 1 for wing stiffness of 3 GPa. Threshold is varied between 0.05 and 0.7. (The normalization constant is only important insofar as it sets the scale of the threshold parameter. The range of thresholds we test therefore represent about 5–70% of the maximum filtered stimulus.)

We then generate spikes probabilistically from the output of the linear-nonlinear encoding. The sensor spikes if the probability of firing exceeds a random draw from a standard uniform distribution. We manually impose an absolute refractory period of 15 ms between spikes. This is not intended to represent the actual absolute refractory period of mechanosensors, but rather to empirically match observations from previous experimental work that each sensor fires only 1–2 spikes per wingbeat [75, 79]. For each data set simulated from the Euler-Lagrange model, we generate 10 sets of spiking responses, for a total of 1,500 data points per optimization (75 wingbeats per Euler-Lagrange simulation, 10 sets of spikes generated for each simulation, for both flapping only and flapping with rotation). Noise in the spike generation step dominates noise in the Euler-Lagrange simulations, so similar results would be expected for shorter Euler-Lagrange simulations and more repetitions of spiking responses generated for each simulation.

## Sensor optimization

Our objective in sensor optimization is to determine the placement of a small number of neural-inspired sensors which can be used to determine whether or not the animal is rotating. We simulate spiking data as described above for two cases: one where a wing is flapping, and one where a wing is flapping and rotating (in either the yaw or pitch axis). For both of these cases, we determine the time to first spike within each wingbeat with 0.1 ms precision and use only this information to classify the data. For wingbeats where no spike is elicited, we designate the spike time as zero, though results are unchanged if we instead designate the spike time as a time longer than the wingbeat duration (e.g., 400 ms spike time compared to 40 ms period). Data are standardized in this optimization step, but the original (non-standardized) data are used to evaluate accuracy.

To determine optimal sensor locations, we use a previously developed method called *sparse sensor placement optimization for classification* (SSPOC) [80]. This method first uses dimensionality reduction (principal component analysis, in our case) to find a lower-dimensional subspace that captures important features of the data. We then use a linear discriminant analysis (LDA) to find the projection vector $w$ that maximally separates the classes of our data in this subspace. Finally, we use elastic net regularization to solve for a sparse set of sensors $s$ that can reconstruct the projection vector $w$. For two-way classification (flapping only, rotation about a single axis), we solve:

$$s = \arg\min_{s'} \left[\lambda \|s'\|_1 + (1-\lambda)\|s'\|_2\right] \qquad \text{subject to } \Psi^T s' = w, \tag{1}$$

where $s$ is a vector of sensor weights ($n$x1, with many near-zero entries), $\Psi$ is the low-dimensional basis ($n$x$m$, $m < n$), $w$ is the projection vector in the low-dimensional subspace ($m$x1), and $\lambda$ determines the balance between $L_1$ and $L_2$ regularization. We set $m = 3$ and $\lambda = 0.9$.

We also perform a four-way classification of: (1) flapping only, (2) rotation in yaw, (3) rotation in pitch, and (4) rotation in roll. In this case, we perform the same elastic net regularization, subject to a different constraint:

$$s = \arg\min_{s'} \left[\lambda \|s'\|_1 + (1-\lambda)\|s'\|_2\right] \qquad \text{subject to } \Psi^T s' - w \leq \epsilon, \tag{2}$$

where $\epsilon = 10^{-6}$.

We use the cvx package to solve this optimization problem (http://cvxr.com/cvx/) [81, 82].

## Performance evaluation

90% of the data is used as a training set for the optimization, and 10% is held out as test data to evaluate accuracy. For straightforward comparison across conditions, we consistently use the top 10 sensors (i.e., sensors with the largest weights in $s$) to assess classification accuracy. We find that 10 sensors are typically enough to achieve near-peak accuracy (Fig 5A, 5B and 5C) without including a large number of extraneous sensors. LDA is again used to find the best projection vector $w_c$ for the non-standardized test data for only the top 10 sensors. For two-way classification, a decision boundary is drawn at the mean of the two condition centroids. For four-way classification, we classify individual points according to the nearest centroid. (Note that the results will be the same whether or not the test data set is standardized. We choose to calculate accuracy based on the non-standardized data to highlight the fact that a linear decoding scheme can be used to read out information from the original spike-timing data.)

## Supporting information

**S1 Appendix. Further examination of factors affecting optimal sensor locations and sensing performance.**
(PDF)

## Acknowledgments

We would like to thank Thomas Mohren and Michelle Hickner for helpful discussions that shaped this project and for contributions to code development.

## Author Contributions

**Conceptualization:** Alison I. Weber, Thomas L. Daniel, Bingni W. Brunton.

**Investigation:** Alison I. Weber, Thomas L. Daniel, Bingni W. Brunton.

**Methodology:** Alison I. Weber, Thomas L. Daniel, Bingni W. Brunton.

**Software:** Alison I. Weber.

**Supervision:** Thomas L. Daniel, Bingni W. Brunton.

**Writing – original draft:** Alison I. Weber.

**Writing – review & editing:** Alison I. Weber, Thomas L. Daniel, Bingni W. Brunton.

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
