## [Decision Letter · Decision Letter 0]

22 Mar 2021

Dear Dr. Weber,

Thank you very much for submitting your manuscript "Wing structure and neural encoding jointly determine sensing strategies in insect flight" for consideration at PLOS Computational Biology.

As with all papers reviewed by the journal, your manuscript was reviewed by members of the editorial board and by several independent reviewers. In light of the reviews (below this email), we would like to invite the resubmission of a significantly-revised version that takes into account the reviewers' comments.

Specifically, two of the reviewers consider that roll (as well as yaw and pitch) should be addressed in your approach. They also note potential limitations with the assumptions in the stiffness model, and that more explicitly connecting this to experimental data (or more clearly discussing the limitations of the model associated with the necessary assumptions) would strengthen the paper. Please see their further detailed suggestions below.

We cannot make any decision about publication until we have seen the revised manuscript and your response to the reviewers' comments. Your revised manuscript is also likely to be sent to reviewers for further evaluation.

Sincerely,

Barbara Webb

Associate Editor

PLOS Computational Biology

Lyle Graham

Deputy Editor

PLOS Computational Biology

Reviewer's Responses to Questions

**Comments to the Authors:**

Reviewer #1: The manuscript by Weber et al. used a numerical approach to study the effect of wing stiffness, neural encoding and strain-sensor position on the ability to distinguish between body rotation and flapping using the strain sensors data. The wing is flapped and parameterized as a hawkmoth wing and the analysis of the strain data is bio-(neural)-inspired. The manuscript is well written. The modelling approach is valid and based on empirical data, the work described is novel and the topic – effect of wing mechanical properties on sensing – is fascinating. For these reasons I don't have a problem with publication, but I do have a fundamental issue with the basic approach.

As the authors rightfully mention, this work is a first step towards understanding how wing structure determine rotation sensory (line 183). As a first step, I see the value of performing the analysis on a very simple shape (a flapped plate, lines 287-292) to isolate parameters (wing stiffness), but I resent the attempt to portray this shape as a “computational model of an insect wing” (line 185). The thing is, that in addition to having veins and membrane the insect wing has non uniform thickness, a different planform (particularly relevant for the sensors at the wing base) and non-uniform stiffness. The SI section re-analyzed the classification with chord-wise strain and gave quite different results (mainly for Yaw) but most of the insect wing-veins are neither parallel to the span or the chord. While some of these limitations are acknowledged in the Discussion, the question remains concerning to the relevance of the simulation results (sensor position, minimal number of sensors, accuracy) to insect flight. The results are more of a ‘proof of concept’ of an idea rather than describing sensing in insects.

Minor comments

Is there a particular reason why roll is not considered or even mentioned?

Line 101 – with 10 sensors… where in Fig 2 or elsewhere is the support for the number of sensors? I had to reach a later section in the paper to understand that these are the 10 best sensors. Maybe revise to “with as little as 10 sensors”

Fig 1A, 3B, 4B sensor location. To prevent doubt please indicate the tip and base of the wings and the leading and trailing edges.

The format in the references list is not uniform (see for example #10)

Reviewer #2: In this manuscript, the authors build upon previous work on the dual role of the wing as an actuator and a sensor. They do this by using computational models to investigate the effect of wing stiffness on sensing body rotations. The core aspects of the methodology were developed in previous papers (Euler-Lagrange flapping wing model-Eberle et al., 2015; optimal sensor locations-Mohren et al., 2018). Here, the authors vary the wing stiffness of the wing model while subjecting it to yaw and pitch body rotations and determine optimal sensor locations for sensing these body rotations. They then quantify the sensors' ability to classify these body rotations and test their robustness to external disturbances.

Overall, I felt the approach to be very interesting and to have a lot of potential to shed light on the role of the wing in sensing. Investigating the role of wing-stiffness and comparing optimal locations for sensing different body rotations are concrete steps towards this goal. However, I feel that the ideas that form the core of the manuscript were not adequately developed (reasons given below). My concerns are all potentially addressable and are aimed at improving the manuscript.

1. Key idea 1: wing stiffness. Because stiffness varies across flying insects, how this property effects the strains sensed by the sensors is of crucial importance. The stiffness range tested in the manuscript was centered around the stiffness of a hawkmoth's wing. However, it was not clear what determined its boundaries and how it related to the diversity of wing stiffness seen in insects. Because of this, I felt a disconnect between the simulations and its implications on "sensing strategies in insects". Bridging this disconnect would improve the manuscript by bringing out the key idea clearly (and maybe provide testable predictions). Here are a few possible starting points to bridging this gap:

- Use a wing stiffness range that is heavily based on experimental data; something similar to the luminance scale shown in Fig. 1A in Sponberg et al., 2015 and the range occupied by hawkmoths.

- It might be possible to obtain high accuracy for different stiffness values by varying the neural threshold (at least for the wing stiffness factor below 1). I felt this to be an interesting aspect worth investigating further, especially in the context of using high neural threshold as way of dealing with low stiffness factor wings.

- Spike timing precision in the context of stiffer wings (Line 103-111) is another interesting aspect that can be investigated, especially in the context of fast flapping insects with stiff wings. I feel that this is should be developed further.

- I felt that there is a need to compare the model-based optimal sensor positions for different wing stiffnesses with those seen in insects. Do the simulations suggest a shift in strategies? An explicit discussion on this aspect would improve the manuscript.

2. Key idea 2: sensor positions for different axis body rotations. Because these sensors groups need to sense body rotations along multiple axes, it was interesting to see optimal sensor positions being computed for yaw and pitch. I was a bit surprised that roll was not added and think it should be included. In addition, I felt that there is a need to pool these sensor positions and come up with a minimum group of sensors needed to sense yaw, pitch, and roll. The supplementary had a short note on the simultaneous classification of yaw and pitch, which should be added to the main results and elaborated a bit more as it suggests that accurate classification requires different groups of sensors. Perhaps the authors could add a note on how three groups of optimal sensors, tuned for yaw, pitch, and roll, respectively, would respond to a perturbation along an intermediate axis. This is important because active body rotations might occur along these axes but external perturbation-induced rotations need not.

3. Caveat of the methodology: the methodology used to determine sensor locations is optimized for classification, that is, to determine the presence or absence of a particular body rotation. Mohren et al., 2018 used the methodology from an engineering perspective and a bio-inspired approach. However, in this manuscript, this methodology is used to better understand wing-based sensing strategies. I feel that there is a need to talk about the caveat of this approach, especially the assumptions and limitations, in the context of insect flight.

Line 20-24: "This … those features." I found the sentence to be too long. I had to read it a few times before I was able to grasp the meaning. I recommend breaking it down into smaller sentences.

Line 98-102: "This results … ~75% accuracy." This difference in spike timing should be shown clearly, with a linked figure.

Fig. 2: The difference in the green shades used for rotation and no rotation is hard to perceive.

Line 109-111: "Lower precision … approaching 100%." This line seems to hint towards something interesting. It needs to be elaborated upon, maybe as a discussion point.

Fig 3, 4: The color difference between dots is not perceptible for certain combinations. Maybe add numbers as well?

Line 138-141: "When the … strategies emerge." An interesting point that needs to be elaborated upon, maybe in the discussion (refer to point 1).

Line 150-152: "First, classification … stiffest wings." I wonder how this changes when one uses two wings to classify rotation instead of one. Pitch/roll related effects, at least with halteres, are ambiguous when one uses information only from one haltere. Adding two wings to the simulation is something the authors should consider. It adds a lot more work to the revision, but I feel it will give more biologically relevant insights.

Line 152-155: "Second, optimal … wing base." Needs to be elaborated in the discussion (refer to point 2).

Discussion: In general, I found the discussion to be too broad and not adequately linked to the results. Some sections were redundant with respect to points raised in the introduction (e.g. body structure transforms sensory inputs). I feel that there is a need to flush out the discussion and link it to the broader context, which in this case is the diversity of wing stiffness and sensory positions seen in insects (refer to point 1, 2). In addition, there is a need to explicitly state the caveats of the methodology (refer to point 3).

Reviewer #3: The manuscript explored the combined effect of structural stiffness and neural encoding on the optimal sensing of flapping flight, which is a novel and interesting topic. The reviewer thinks the quality of the manuscript meets the standard of publishing with PLOS Computational Biology. The reviewer also has several questions as follows,

The results show that when the stiffness and the neural threshold change, the optimal sensor locations are clustered either at the tip or the base of the wing during yawing motion (Fig. 3B). The reviewer is curious about what causes this abrupt change of the optimal locations from one side of the wing to another. What might be the underlying physics that may explain this change? The reviewer is also curious about whether there is a scenario when the optimal locations of the sensor are located away from the four corners of the rectangular wing. A finer parametric study of the stiffness‒neural threshold space may help answer this question.

Can the authors try to explain why in the pitching motion, the accuracy is much reduced and the optimal sensor locations are less clustered when compared with those of the yawing motion?

The rotations in this manuscript (yaw and pitch) are performed at constant angular speed. Since insects often perform maneuvers with accelerations, will the current methods be able to access the effect of angular accelerations?

**Have all data underlying the figures and results presented in the manuscript been provided?**

Reviewer #1: None

Reviewer #2: None

Reviewer #3: Yes

PLOS authors have the option to publish the peer review history of their article (what does this mean?). If published, this will include your full peer review and any attached files.

Reviewer #1: No

Reviewer #2: No

Reviewer #3: No
---

## [Decision Letter · Decision Letter 1]

18 Jun 2021

Dear Dr. Weber,

We are pleased to inform you that your manuscript 'Wing structure and neural encoding jointly determine sensing strategies in insect flight' has been provisionally accepted for publication in PLOS Computational Biology.

You may want to take into consideration the suggestions below of reviewer 2 when preparing the final version, though we leave that up to you.

Best regards,

Barbara Webb

Associate Editor

PLOS Computational Biology

Lyle Graham

Deputy Editor

PLOS Computational Biology

Reviewer's Responses to Questions

**Comments to the Authors:**

Reviewer #1: Thank you for making the changes. I have no further questions or comments. The revised manuscript provides a more complete picture and I believe it will provide an important contribution to the effort of deciphering how insects sense their rotation using wing mechanosensors.

Reviewer #2: I feel that the authors have adequately addressed all my concerns/suggestions/comments. I really appreciate their efforts to address my concerns about roll detection, rotation along an intermediate axis and sensing with both wings. I also feel that the discussion is now pretty fleshed out. Overall, it is clear that authors have put in substantial effort to strengthen the paper based on the comments from reviewers.

There are still a few things I am curious about. These are minor concerns which need not be addressed (but would be marvelous if they are).

1. I was a bit surprised to see that the two wing sensing strategy was applied to only yaw and not to pitch and roll detection. I made this suggestion because pitch detection is worse than yaw (and now roll) detections and using two wings might boost performance for pitch tasks.

2. I realize I was not very clear about my suggestion on perturbations along an intermediate axis. What I was trying to get at was the generality of sensors optimized for yaw, pitch and roll to detect rotations along an intermediate, unoptimized axis. That is, when groups of sensors, each optimized for yaw, pitch and roll, are put together on the same wing, would these automatically confer the ability to detect rotations along intermediate axes? Would the dutch-roll activate the yaw and roll group of sensors?

The dutch roll optimizations and the 4-way classification does indirectly test the same thing - the results do show the use of combinations of single-axis strategies. I feel that it might be better to test this explicitly as it allows the results to be generalized further.

3. I quite like the new constrained sensor location analysis (Supp Fig 7). It led me to wonder if one could look at the sensor positions for yaw, pitch, roll and find a common base sensor location set (say 50 locations). Could one then show that by constraining the sensor locations to these regions, one could still accurately classify flapping vs flapping + rotation along any axis? This might strengthen the argument that campaniform sensillae located at the base can detect rotations (Discussion point: Perhaps the most striking commonality across diverse taxa is the relative abundance of campaniform sensillae near the wing base [47-49, 51, 53-55].).

Reviewer #3: The authors have satisfactorily responded to all my questions and the manuscript has been substantially improved. Therefore, I recommend the acceptance of this manuscript to be published.

**Have the authors made all data and (if applicable) computational code underlying the findings in their manuscript fully available?**

Reviewer #1: Yes

Reviewer #2: None

Reviewer #3: None

PLOS authors have the option to publish the peer review history of their article (what does this mean?). If published, this will include your full peer review and any attached files.

Reviewer #1: No

Reviewer #2: **Yes: **Dinesh Natesan

Reviewer #3: No

---

## [Editor Report · Acceptance letter]

19 Jul 2021

PCOMPBIOL-D-21-00114R1 

Wing structure and neural encoding jointly determine sensing strategies in insect flight

Dear Dr Weber,

I am pleased to inform you that your manuscript has been formally accepted for publication in PLOS Computational Biology. Your manuscript is now with our production department and you will be notified of the publication date in due course.

With kind regards,

Zsofi Zombor
